# Health Status Classification for Cows Using Machine Learning and Data Management on AWS Cloud

**DOI:** 10.3390/ani13203254

**Published:** 2023-10-18

**Authors:** Kristina Dineva, Tatiana Atanasova

**Affiliations:** Institute of Information and Communication Technologies, Bulgarian Academy of Sciences, Acad. G. Bonchev Str., Bl. 2, 1113 Sofia, Bulgaria; tatiana.atanasova@iict.bas.bg

**Keywords:** dairy cows, data analysis, data modeling, data integration, random forest classifier (RFC), health status classification, model integration, Amazon Web Services (AWS)

## Abstract

**Simple Summary:**

Digital transformation in modern farms is triggered by the development of technologies. It allows the monitoring of livestock and evaluation of animal welfare by using data from an increasing number of sensors and IoT devices. This research supports farmers with information on cow health status classification based on the non-invasive IoT sensors and information on the micro- and macroenvironment of the cow. The collected data from various sources are processed, modeled, and integrated following the proposed workflow. Several machine learning (ML) models are trained and tested to classify cow health status into three categories. The results are visualized for the farmer’s use. This approach is different from other studies because we investigate how microenvironments, macroenvironments, and cow’s information influences the cow health status and whether a combination of these can support and increase accuracy and reliability in the classification process. It provides a practical solution for monitoring large farms, particularly suitable for the livestock industry.

**Abstract:**

The health and welfare of livestock are significant for ensuring the sustainability and profitability of the agricultural industry. Addressing efficient ways to monitor and report the health status of individual cows is critical to prevent outbreaks and maintain herd productivity. The purpose of the study is to develop a machine learning (ML) model to classify the health status of milk cows into three categories. In this research, data are collected from existing non-invasive IoT devices and tools in a dairy farm, monitoring the micro- and macroenvironment of the cow in combination with particular information on age, days in milk, lactation, and more. A workflow of various data-processing methods is systematized and presented to create a complete, efficient, and reusable roadmap for data processing, modeling, and real-world integration. Following the proposed workflow, the data were treated, and five different ML algorithms were trained and tested to select the most descriptive one to monitor the health status of individual cows. The highest result for health status assessment is obtained by random forest classifier (RFC) with an accuracy of 0.959, recall of 0.954, and precision of 0.97. To increase the security, speed, and reliability of the work process, a cloud architecture of services is presented to integrate the trained model as an additional functionality in the Amazon Web Services (AWS) environment. The classification results of the ML model are visualized in a newly created interface in the client application.

## 1. Introduction

Animal health and welfare are of great importance to the agricultural industry and are an integral part of the European Union’s new Farm to Fork (F2F) strategy [1]. The health and welfare of food-supplying animals depend largely on how they are managed by humans [2]. Currently, animal husbandry is expanding with large-scale and intensive development, with a prominent tendency to create and build farms raising many animals. The ability to timely detect animal health issues in large farms is essential to prevent epidemics and minimize economic damage caused by reduced and poor-quality production. Traditional methods of caring for large numbers of animals are often labor-intensive and time-consuming [3,4,5,6]. In large farms, this leads to insufficient care and attention [7] and not meeting the needs of the animals, which in turn leads to the occurrence of diseases and, as consequences, deterioration of the quality of production.

The digital transformation in livestock management allows large amounts of information to be stored and used to make decisions based on quantitative and qualitative analytical results [3]. Internet of Things (IoT) devices and RFID-based electronic identification [8] are employed intensively in smart farming to collect a huge amount of heterogeneous data used to provide several valuable insights. Data from sensors processed with artificial intelligence (AI) and machine learning (ML) have been used in applications on dairy farms recently [9,10]. For example, K-nearest neighbors (KNN), random forest (RF), and extreme boosting algorithm (XGBoost) were applied to classify unitary behaviors of 10 dairy cows [4]. Machine learning techniques like logistic regression, Gaussian naïve Bayes, and random forest were used to predict the accuracy of the scoring system to evaluate the severity of heat stress on dairy cows [11]. A random forest-based model for the assessment of the daily milk yield at the single cow level was tested in [12]. The extreme gradient boosting algorithm (XGB) was performed to classify the features of rumination and eating behaviors [13].

The investigation in [14] noted that most animal science research led by sensors and ML models has been focused on data collection, processing, evaluation, and analysis in the fields of animal behavior detection, disease monitoring, growth assessment, and environmental monitoring at the experimental stage. Several automatic methods for detecting indicators of reduced welfare have been developed [15] recently. However, current digital methods for livestock diagnosis are mostly based on the analysis of only one type of information on livestock characteristics. This affected the accuracy of the results of the diagnostic system which were not accepted as conclusive [16]. In addition, there are still some problems with current digital methods of livestock analysis, such as the poor ability to generalize the diagnosis and the poor ability to prevent interference, which restrict their promotion and application.

The literature review provided in [17] on research devoted to the construction of health detection models shows clear gaps in sensor information integration and decision support systems, as well as predictive methods by identifying low-resilience animals. At the same time, many robotic farms still rely on veterinarians to assess welfare and treat production-related illnesses or complications [18].

Several challenges to achieving classifying livestock health should be mentioned:Data variability—Cow health data can be highly variable due to factors such as age, breed, environment, and disease history [7,19]. The resistance threshold of each cow varies depending on the phase of its life cycle. This makes it difficult to establish clear and consistent patterns that accurately reflect the health status of the animal. It is crucial to consider the individual cow’s traits.Complex interactions between health factors—The health of livestock is influenced by a variety of interconnected factors [4,6,7], such as genetics, feeding type and frequency, environmental conditions, and local infestations. This makes it challenging to isolate the impact of individual factors on overall health status.Subjectivity in observations—Assessment of animal health is often based on subjective observations made by farmers or veterinarians, which often leads to biases and inconsistencies in the data.Lack of data—Automated data collection equipment may not always be readily available on farms, which is why some farmers still record their data in notebooks.Reliable and secure equipment—Building trust in automated systems takes time and requires consistent proof of their reliability. In order to reduce human intervention in animal husbandry and support monitoring processes, it is crucial to rely on existing equipment and ensure that data are transmitted securely and in a timely manner.

Considering the challenges, the main objective of this research was to create a combined dataset including data on micro- and macroenvironments and individual information for each cow (health diary). To obtain consistent data without adding stress to the animals, data from already incorporated IoT devices and tools were used for the purpose of this research. The collected data were analyzed and transformed to be suitable for training a machine learning algorithm to obtain a classification ML model to identify the health status of cows in three categories—healthy, unhealthy, and suspect. Due to the need to apply different steps, techniques, and methods for data processing, a systematized workflow has been created that increases the clarity and facilitates the repeatability of the experiments, so that it can serve as a starting point for future research. To automate the process, a cloud workflow architecture has been established to ensure reliable, timely, and secure transmission, storage, processing of data, and classification results.

The rest of this paper is organized as follows. Section 2 introduces the materials and methods used to develop a custom multiclass classification model. The used IoT tools and data characteristics are explained. A workflow for IoT data processing, modeling, and integrating is proposed. The workflow is explained in detail and followed strictly to obtain reliable and reproducible outcomes. Also, a model integration design diagram for new data and ML model incorporation in a cloud environment using serverless cloud services is presented. In Section 3, a description and visualization of the experimental results are provided. In Section 4, the obtained results are discussed, and limitations are highlighted. Future work directions are mentioned. Finally, Section 5 concludes the paper.

## 2. Materials and Methods

### 2.1. Data Collection

Data in this study are collected only from already mounted IoT devices and tools (Figure 1) in a dairy farm. The design structure, layout, and equipment of a barn can be found in Appendix A. Figure 1a,b illustrate two components of a single system. This system automates the process of extracting milk, measures the quantity of milk for each cow, and transmits these data to the system. Figure 1c shows an IoT device attached to the cow’s neck, which is used for microclimate data collection and cow identification. Figure 1d shows the primary device used to set the needed temperature and humidity thresholds. Each sector of the barn contains IoT devices for control. Additionally, fans and water sprinklers are included and installed at the top of each sector. The used approach represents a non-invasive method for data collection.

A cow’s microclimate is influenced by both internal and external factors. External factors consist of the general climate conditions, size of the barn, number of cows in a group, and the cooling/heating equipment used. The weather station collects macroclimate data during a data collection period that lasts for 12 months and includes all four seasons. Internal factors refer to various aspects affecting a cow’s body temperature. For instance, feeding frequencies and the time of day play a significant role as the cow’s body temperature is generally lower in the morning after rest and higher at night after a day of muscle activity. Additionally, milking, hormonal fluctuations, and other factors can cause a change in temperature.

Data were collected on approximately 120 Holstein dairy cows (black and white) located on a farm in central Bulgaria. The area experiences a continental climate.

This research combines data obtained from IoT tools, as well as data from a weather station located near the farm, and details about each cow’s individual health recorded manually by the farm’s veterinarians in Excel (version 16.57) spreadsheets (health diary). The IoT tools and weather station send data at different time intervals. The data are saved and synchronized over time. Data for up to four daily measurements are collected to create a dataset. However, the data for each cow were collected for a different time interval based on when they entered or exited the herd, so there are different records for each cow.

To organize the data, a table was created with columns for the cow’s ID, the date and time of the measurement, and the measurement values. Then, grouping by the cow’s ID and the specific period is performed. The data contain a total of 2,285,665 values, which are organized into 53,155 rows and 43 columns.

### 2.2. Workflow

The extracted data are in a raw format, so it is necessary to analyze and format them into a suitable form for developing a custom classification ML model. The proposed systemized workflow shown in Figure 2 combines methods, approaches, processes, and good practices to answer the question of how to process and model data obtained from IoT devices in combination with other data sources and also how to integrate trained machine learning models into production environments. It is viable for IoT systems which collect data that need to be analyzed, processed, modeled, and integrated.

The workflow consists of four stages with the following steps:1.First stage: Problem statement
Problem definition/hypothesis statement.
2.Second stage: Data preparation
Data acquisition.Exploratory data analysis (EDA).Data pre-processing.
3.Third stage: Data modeling
Selection of proper machine learning algorithm.Hyperparameter tuning.Model training and testing.Model evaluation.
4.Fourth stage: Integration layer
Feed data.Data pre-processing.Prediction results.Model monitoring.Model validation.Data visualization and interpretation.Model maintenance.

#### 2.2.1. Problem Statement

The approach being considered is to gather information on the micro- and macroenvironment, as well as the unique characteristics of each cow, considering important health indicators studied and outlined in [15]. The data must then be processed to create a dataset suitable for training a multiclass classification machine learning model that can accurately classify the animal’s health status based on the animal’s specifics and its environment. This leads to improved early detection of potential health problems and prevention of disease outbreaks in the livestock industry.

#### 2.2.2. Data Preparation

##### Data Acquisition

Although IoT devices can gather a vast amount of data, not all of them are useful in determining the health status of animals. In fact, including irrelevant or redundant data in the classification model can make it more complex and decrease its overall performance. To avoid these issues, we select 16 specific characteristics, which include both IoT and individual cow’s data. These features are presented in Table 1.

Data collected by cow’s neck IoT devices with sensors are temperature, humidity, and animal identification. Data collected by other IoT tools are registration number (cow ID), daily milk yield, average milking days (DIM), temperature, humidity, wind direction, cloud level, and rain quantity. Data extracted from the cow’s diary are registration number (cow ID), age, gender, lactation, group, and health status.

The data obtained from the health diary for the health status column are utilized as the primary training data for the ML model. The values in the health column are divided into three categories: healthy, unhealthy, and suspect. To be classified as healthy, the vet states that the cow is not in any discomfort. The “unhealthy” group includes animals with conditions caused by cold or heat stress. The “suspect” group indicates the presence of sufficient conditions for the occurrence of a disease state in the animal, but it has not yet manifested itself. To construct the dataset used to train the algorithm, the veterinarian manually enters data into a separate temporary column in the cow’s diary labeled “health status”. This column specifically indicates cows that are either unhealthy or suspect. Once trained, the algorithm can use historical data to detect patterns of when to expect decreases or increases in milk yield, which can be natural phenomena during different stages of a cow’s development. It can also identify the micro- and macroenvironments in which unfavorable conditions for cows can occur. By discovering and learning patterns related to the desired outcome—the health status—the algorithm automatically classifies the cow’s health status.

##### Exploratory Data Analysis

EDA is an essential step in the data preparation stage as it helps to ensure the data quality (format, consistency, accuracy, etc.) is appropriate and provides insights that can guide feature selection and engineering, model selection, and hyperparameter tuning.

In this study, EDA is divided into two parts—firstly a univariate analysis and then a bivariate analysis.

Univariate analysis is a statistical analysis technique that involves examining one characteristic at a time to describe the data and identify patterns or trends in the data.

Bivariate analysis is a statistical analysis technique that focuses on analyzing the linear relationship between every two features in a dataset. It is used to understand the nature and strength of the relationship between features and inform feature selection or engineering.

This way, the basic structure of the data, the distribution, and the characteristics of each variable are discovered. Also, outliers, missing values, highly correlated features, and others are identified. Such values should be handled appropriately to avoid model bias.

##### Data Pre-Processing

The data-pre-processing step is responsible for the data being cleaned, transformed, balanced, and formatted for better machine learning algorithm efficiency.

In this study, the following data pre-processing is performed:*Cast objects to a specified data type.* This is the process of converting the variables (values in one column) into the most appropriate data type. Performing this operation allows grouping the categorical values and checking the distribution, as well as deriving statistical measures from the numerical values.*Process missing values*. In the context of IoT data, null, Na, Nan, and NaT values are particularly problematic because they indicate a sensor outage, loss of data in communication, or equipment problem. Null values are removed from this study because they are strongly suspected to be problematic rather than natural. This ensures that the resulting models are built on complete and accurate data, thereby improving the overall quality of the analysis.*Process outliers*. Outliers are observations that lie at an abnormal distance from other values in a random sample from a population. Both missing values and outliers in IoT data need to be further investigated to determine whether they are a random phenomenon or if there is any cyclicality of their occurrence. For this study, the interquartile range (IQR) is calculated, and the handled outliers were treated by the clipping function [20] instead of being deleted because the data contain important information that could show clear signs of the presence of diseases, which directly affect the classification of the animal’s health status.*Bucketizing (binning).* Bucketization of continuous features is a technique used after the data analysis is performed or after ML error analysis. Simplifying complex data by reducing the number of unique values makes the data easier to understand. It also improves the performance of models such as decision trees and random forests.*Balancing data*. Data are unbalanced when there is a significant difference in the number of instances between categorical classes of data. This can cause ML models to prefer the majority class and incorrectly classify instances from the minority class. In this study, there are three equally important classes that need to be correctly classified. To address the imbalance, the synthetic minority over-sampling technique (SMOTE) [21] was utilized in the minority class. This technique synthesizes new examples to balance the number of instances across all classes.*Categorical encoding*. Label encoding is a technique used to convert categorical data into numerical. The label-encoding technique used in this study is one-hot encoding. It involves converting categorical variables into a set of binary columns, where each column represents a unique category in the original variable. The column for a particular category is set to *1* if that category is present in the observation and *0* if it is not present.*Data standardization*. IoT devices gather various sensor data with varying measurement types and scales. Data standardization scales each input variable separately by subtracting the mean and dividing by the standard deviation to shift the distribution to have a mean of *zero* and a standard deviation of *one*. The type of standardization used in this study is standard scaler because it preserves the variance of the data.

Overall, data pre-processing is a crucial step because it ensures the input data are of high quality, properly formatted, and suitable for the machine learning algorithm.

#### 2.2.3. Modeling

To categorize complex patterns and relationships within a dataset, machine learning algorithms are utilized. Prior to training and testing multiple ML algorithms, the data are split into training and test sets using stratified sampling to ensure a similar balance of target class weight. The training data comprise 70% of the total data, while the testing data use the remaining 30%. Grid search is then employed to discover the optimal parameters for each algorithm, and all machine learning algorithms are tested with *k-fold = 10* to guarantee the best hyperparameters are determined for each model.

##### Selection of Proper Machine Learning Algorithm

Five machine learning algorithms [22], described in Table 2, are selected to classify cow health status into three categories—healthy, unhealthy, and suspect. The same dataset is used for training and testing the five algorithms to fairly compare the classification performance.

##### Hyperparameter Tuning

Finding the optimal model hyperparameter values improves accuracy, reduces the risk of overfitting or underfitting, and makes the model more robust to different types of input data. It allows model behavior to be fine-tuned to better solve the stated problem, leading to better classification and better results. In this study, the grid search CV method for performing hyperparameter optimization is used for the five models considered. Different parameters and a different range of values are set for each model. The best combination of parameters for each model is sought. The goal is to train and evaluate each of these models using cross-validation and to choose the model with the best performance metrics. The obtained hyperparameters are used to control the learning process that guarantees the repeatability of results for given input data and hyperparameter settings.

##### Model Evaluation

Evaluating the performance of a multiclassification model is essential to assess its effectiveness. To effectively evaluate the models, a confusion matrix is constructed that summarizes the number of correct and incorrect classifications made by a model for each of the three classes. The confusion matrix provides information on the true positive (TP), true negative (TN), false positive (FP), and false negative (FN) rates. Model accuracy (1), precision (2), recall (3), and F1-score are calculated by the following formulas:Accuracy = (TP + TN)/(TP + FP + TN + FN)(1)
Precision = TP/(TP + FP)(2)
Recall (Sensitivity) = TP/(TP + FN)(3)
F1 score = 2 × Precision × Recall/(Precision + Recall)(4)

Our aim is to identify all actual positive cases and limit the admission of false negatives—type 2 errors. The primary metric used to evaluate model performance is *recall*. It answers the question “How sensitive is the model to detecting positive cases?”. To prevent the misclassification of sick cows as healthy, we need to limit the instances where this occurs. This is known as a false negative result, and it can worsen the disease if the animal is not given proper attention and excluded from the priority group for medical examination. This group for medical examination usually is formed by animals with positive results.

##### Error Analysis

Error analysis helps to identify and discover the causes of errors made by the model. By identifying the types and causes of errors (FP and FN), areas where the model needs improvement can be identified and analyzed. This is important because where and how we make errors have very different real-world costs.

Errors can be caused by poor data quality, mislabeled data, missing data, duplicate data, inconsistent data, lack of important data for our classification, and many others. Per-forming this type of analysis helps to further understand the characteristics of the data and when errors occur. This can lead to more effective feature engineering, data pre-processing, and model selection by understanding exactly where our model is making mistakes. To perform proper error analysis, it is important to carry out the inverse transformation of the values from the standard scalar back into the original data’s original form.

##### Data and Model Integration

The final stage of the proposed workflow is the integration of the trained model with new data in a production environment. The developed model is integrated as additional functionality to an already developed Amazon Web Services (AWS)-based livestock monitoring system explained in [24]. The integration design is presented in Figure 3. It consists of data and machine learning layers for processing data, training ML models, and classifications.

The data layer contains the following groups of services:*AWS Glue*—Serverless data integration service. It consists of crawlers, jobs, triggers, and a data catalogue. It is responsible for data extraction from multiple sources, data preparation, and loading them into the data lake [25].*Data lake*—A data lake is an architectural approach that allows storing data in a centralized repository. It consists of simple storage service (S3) buckets. S3 is a petabyte-scale object store which provides virtually unlimited scalability to store any type of data [26].

The modeling layer contains the following services:*Step Functions*—Serverless orchestration service that automates processes, orchestrates microservices, and creates data and machine learning pipelines [27].*SageMaker*—provides highly scalable and security tools to build, train, test, tune, and deploy custom ML models [28].

The AWS Glue workflow has pre-defined triggers for crawlers and jobs. The crawlers periodically check for newly added raw data that is stored in the S3 buckets. When new data are available, the crawlers add and/or update the tables in the created AWS Glue data catalogue, which serves as a central metadata repository. When a crawler’s run is completed, Python shell jobs are started using Python scripts performing pre-defined pre-processing steps on the data such as organization (tidy data), processing missing values, standardization, and others. After the completion of those jobs, the results are stored for subsequent analysis and ML modeling in an S3 bucket dedicated only to prepared data. The model layer is using AWS Step Functions and AWS SageMaker. AWS Step Functions uses pre-pared data from the data layer S3 bucket for training the ML models and validating the metrics during the training process. When an ML model is in a production-ready state, it is deployed and registered in the AWS SageMaker Model Registry and a model endpoint is created. The model endpoint is called to perform ML tasks. Batch transform can also be triggered three times a day after the milking of the cows is finished and sufficient data from the IoT tools are received.

In the case the results pass the evaluation metrics check, they are added for storage in the S3 bucket and then used for subsequent visualization on-demand using the client app. If the ML results are below a certain threshold, then the model improvement is triggered to improve the ML model.

## 3. Results

### 3.1. Univariate and Bivariate Exploratory Data Analysis

As a result of the univariate analysis of all features, valuable information about the distribution, central tendency, and shape of a variable was obtained.

#### 3.1.1. Univariate Analysis—Numerical Features

The presented histogram in Figure 4 shows the data distribution of numerical features. This gives sufficient information to understand the data patterns and how the values of numerical features are spread out across the range of possible values.

Based on the histogram representation of the data, it is evident that the data points for “Lact.”, “DIM”, “Daily Milk”, and “Age” have a positively skewed distribution. This means that most data points for these variables are concentrated toward the lower end of their ranges, with a tail extending toward higher values. On the other hand, “Humidity” and “Humidity_IoT” have a negatively skewed distribution, indicating that most data points are clustered toward the higher values, and there is a tail extending toward lower values. Meanwhile, the “Rain” data points have a unimodal distribution, which means it has a single peak, while “Temp.”, “Temp_IoT”, “Cloud”, and “Grp.” have a multimodal distribution—this implies that there are multiple significant peaks or modes in the data, possibly indicating different subpopulations or varying underlying processes.

Overall, analyzing the distributions of these columns, we can gain useful knowledge about the dataset’s characteristics. This determines the appropriate data pre-processing and feature engineering techniques, as well as the statistical methods to use. The analysis reveals the variety of different scales, which requires normalization of the data, and the presence of outliers and multiple similar values is also noticeable. The datetime type needs to be separated into year, month, week, and day.

Descriptive statistical analysis is also carried out. It involves calculating and summarizing descriptive statistics such as mean, standard deviation, min value, and 25%, 50%, 75%, and max of value, that are presented in Table 3.

The presented table provides a description of various numerical characteristics. Each row of the table represents a statistical measure, and each column represents a different characteristic. From the analysis, there are missing values in “Lact.”, “DIM”, “Daily Milk”, “Temp_IoT”, and “Humidity_IoT” with 6.9% missing values overall. The presence of outliers was detected, and the variation of values was confirmed.

#### 3.1.2. Univariate Analysis—Category Features

There are three object features in the dataset—“*Health Status*”, “*Wind_dir*”, and “*Gender*”. Figure 5 displays histograms illustrating the distribution of feature classes, revealing significant class imbalances. Substantial disparities are evident within the categories. Figure 5a depicts the distribution of the target feature, “Health Status”, showcasing its imbalance. In Figure 5b, the distribution of “Wind_dir” values is presented. Figure 5c represents the categorical distribution of “Gender” within the dataset.

From the histograms, it is evident that there is a significant imbalance in the unhealthy, healthy, and suspect classes, with a ratio of 9.5%: 63.1%: 27.4%. To address this issue, it is essential to balance the three classes. Additionally, bucketing is necessary for the “Wind_dir” feature, as it has several imbalanced classes with minimal observations. Moreover, the “Gender” feature is highly imbalanced, in fact only female cows are the focus of this study. Therefore, it is mandatory to remove rows that contain data for males.

#### 3.1.3. Bivariate Analysis

Bivariate analysis examines how a feature is related to another feature in the dataset. Figure 6 shows a correlation heatmap that is used to quantify the degree of association between two features and to identify any patterns in the relationship using the Pearson method [29]. Pearson correlation coefficient is a statistical measure of the linear relationship between two variables. The correlation coefficient can range from −1 to 1.

From the above correlation heatmap, the following information on the strength of relationships between numerical variables could be obtained:The variables “Temp.” and “Temp_IoT” have a strong positive correlation with r = 0.97.The variables “Age” and “Lact.”, “Month” and “Temp.”, “Humidity” and “Humidity_IoT”, “Temp_IoT” and “Month” have a positive correlation with values of, respectively, r = 0.54, 0.34, 0.56, and 0.28.The variables “DIM” and “Daily Milk”, “Temp.” and “Humidity”, “Temp.” and “Cloud”, and “Temp_IoT” and “Humidity_IoT” have a negative correlation with values of, respectively, r = −0.41, −0.57, −0.18, and −0.27.

In addition to the results of correlation analysis, which identifies relationships between variables, statistical significance analysis is performed to help determine whether observed relationships between variables in a dataset are likely to be real or may have occurred by chance. The statistical significance results between the variables are presented in Appendix B. Correlation analysis and statistical significance analysis work together to provide a comprehensive understanding of the relationships in the data. Moreover, the data are segmented into two categories, representing healthy and unhealthy cows, to gain a clearer understanding of the features’ correlations. The findings can be found in Appendix C.

### 3.2. Data Pre-Processing

The performed EDA indicates a need for transformation of the values in the columns. This transformation includes casting objects to a specified data type, processing null values, processing outliers, bucketing, balancing data, categorical encoding, data standardization, and principal component analysis.


*Object casting*


As a result of casting objects to a specified data type, the dataset has the following distribution: three columns with categorical type, six columns with integer type, six columns with float data type, and a column with date–time type.


*Missing value processing*


As a result of handling the missing values, the dataset is reduced by 6.93% using the imputation method, where only the missing values from the rows are reduced without changing the number of columns. Also, 0.2% of “Gender” column rows were removed that contain data for males.


*Outlier processing*


When handling outliers using the truncating method, the clipping function is per-formed on the detected outliers. As a result, a reduction of the right end values in the “Lact.”, “DIM”, “Daily Milk”, and “Age” characteristics and a reduction of the left end values in the “Humidity” column was performed.


*Bucketing (binning)*


As a result of bucketing, a new column was created—“Daily_bin”—which combines the values from the “Daily Milk” column into three balanced groups. Four new columns were created that combined values from the “Status +” column. Also, the “Wind_dir” column was bucketed into five groups.

#### Balancing

During EDA, it was found that the ratio of classes in the target column was highly unbalanced. After executing the null value-processing step, the ratio of the three classes in the target column changes as follows: healthy (0): 50.5%, suspect (1): 38.2%, and un-healthy (2): 11.4%. Despite reducing the imbalance between the columns, a balancing algorithm is still needed. The synthetic minority over-sampling technique (SMOTE) with sampling strategy {0:0, 1:1.3, 2:4.4} was applied. The algorithm creates a vector between the minority data points and any of its neighbors and places a synthetic point.

In this study, the class “imblearn.over_sampling.SMOTE” from the scikit-learn Python library was used to balance the dataset. As a result, the performance of the model has been enhanced by applying the SMOTE algorithm. The accuracy of the model has in-creased to 0.9598, which is an 11.92% improvement compared to using the unbalanced dataset. When training the random forest classifier with unbalanced data, the model performed well in classifying Classes 0 and 1 with high precision and recall results. How-ever, the model struggled with Class 2. This indicates that the model is biased towards the majority Classes 0 and 1 and is unable to accurately classify the minority Class 2.

In contrast, when training with a balanced dataset, the model improved significantly. The precision and recall metrics for Class 2 have improved to 0.96 and 0.95, respectively. This shows that the model can now classify all three classes more accurately after applying SMOTE.


*Categorical encoding*


The dataset contains three categorical features. A label encoder technique was ap-plied to each of them. As a result, the three columns contain numerical values that represent each class. Accordingly, the classes in the “Healthy” column are numbered as follows: healthy: 0, suspect: 1, and unhealthy: 2. The classes in the “Wind_dir” column are numbered as follows: 0:0, E:1, S:2, N:3, W:4. The classes in the “Gender” column are reduced to one and transformed as follows: female: 0.


*Standardization*


Standard scaler uses feature scaling to standardize features by removing the mean and scaling to unit variance. As a result, the mean value of all features for training and testing (except the target column) is *0* and the standard deviation is *1*.

### 3.3. Data Modeling

The multiclass classification performance shown in Table 4 presents a comparison of five different ML algorithms with different hyperparameter settings reported as the best parameters from the *grid search CV* method. The applied ML algorithms on the cow health status dataset are KNN, Gaussian naïve Bayes, decision tree classifier, XGBoost, and random forest classifier. They were compared according to three indicators: *recall*, *precision*, and *accuracy*. Results vary with each algorithm. The lowest results were obtained by Gaussian naïve Bayes (accuracy 62%), and the highest results were obtained by random forest classifier (accuracy 96%). The results of other three algorithms are similar with less than a 5% difference in accuracy between them: KNN with 82%, decision tree classifier with 83%, and XGBoost with 78%.

In this study, the *recall* metric is the determining factor for algorithm selection. The highest scores on this metric were obtained with KNN and random forest classifier, 73% and 96%, respectively. The difference between the two best results is 23%, which we assume is a large percentage difference for this type of study, therefore random forest classifier performs best in health status classification.

The confusion matrix of the random forest classifier model is presented in Figure 7. The matrix compares the actual target values (*y*-axis) with those classified (*x*-axis) by the machine learning model. The correct classifications of the model are presented in the diagonal, which are 95.98%, and the wrong classifications of the algorithm are presented in the upper left and lower right corners, which are 4.01%. The errors need to be further investigated. Based on the obtained results presented in the matrix, model evaluation metrics are calculated.

The random forest classifier report performance is shown in Table 5. It contains information on the *precision*, *recall*, and *F1-score* metrics for each of the three categories, which are calculated based on values from the presented confusion matrix. The presented report contains results from the testing set. The difference in the results between the training set, 0.9620, and testing set, 0.9598, is within permissible differences, therefore, at this stage, we consider that there are no prerequisites for overfitting the developed model.


*Model Error Analysis*


As a result of the performed model error analysis, it was found that the most errors were made when classifying Class 1—suspect, which was classified as Class 0—healthy. Conversely, Class 0 is classified as Class 1. There are also errors in classifying Class 2—unhealthy, which is most often confused with Class 1. The biggest problem is errors where Class 2 is classified as Class 0. This indicates the presence of false negative results, with these cases comprising 0.71% of the total. After further investigation into the causes of these errors, it was found that they occur at Daily Milk values in the range of 40 to 60 (high levels of daily milk yield). Cow’s age below 500 days and lactation phase over 4 are established as other reasons.

### 3.4. Data and Model Integration

The integrated machine learning model needs to be monitored to evaluate its effectiveness. Monitoring uses AWS CloudWatch metrics to examine SageMaker model performance in near real time. If necessary, retraining or updating the model is performed to ensure the classifications are accurate over time.

The process of validating the developed model lasts up to three months. After this period, we can consider that the classified values are reliable. The results can be visualized and included in the pipelines of other automated processes on the farm.

A client app is created for farmers and vets to validate the classified and actual health status of the animals. It calls on demand the SageMaker model endpoint and receives back the ML classifications. Also, a pipeline is configured that automatically updates the health status classifications in the table when the event is initiated—completion of the milking process and the data arriving in the S3 bucket.

The client app has an interface for easier results visualization and interpretation. It consists of a table and a tooltip with additional information that is shown in Figure 8.

The presented table has four columns containing values that represent individual cows. Each cow is identified by the “Cow Number” column. The “Grp.” column indicates the cow’s group. Also, there is an “Actual Health Status” column displaying the outcomes of a manual inspection conducted by a veterinarian. The veterinarian records details of the performed examinations in the electronic diary. As a result, these three columns are derived from the electronic diary of the cow. The “Predicted Health Status” column in the table shows the results generated by the developed ML model. The aim is to facilitate the comparison of the results, thereby enabling the assessment of the credibility of the model by confirmation or rejection after the duration of the observation. The embedded tooltip becomes active exclusively when hovering over the highlighted line. It presents visual information on additional data obtained from the cow’s electronic diary, such as age, lactation, DIM, and the volume of milk that has been extracted. An additional component is included in the form of a text field for the visual presentation of cow-specific information entered by the veterinarian in the electronic diary. The purpose of introducing the new text column is to collect and visually present details (diagnoses) during the manual veterinary examination of the cow. Collecting data on different types of diagnoses, along with micro- and macroenvironmental data, can provide us with valuable information for future research and can help us improve the current model and predict specific diagnoses in the future. Nevertheless, missing records can be observed in the “Actual Health Status” column of the table. These omissions occur because this cow has not been examined by a veterinarian. Therefore, the categorization performed by the model cannot be validated or rejected due to missing information, for example, for cow 407 in Figure 8. The presence of a significant number of missing values may necessitate an extension of the time frame for model checking. There are two ways that visualization and classification results can be updated. First, farmers/veterinarians can update their health information from the client application. IoT tools send new measurement data to the cloud system. Once the updated data are submitted, the classification model processes them and classifies a health state based on the latest available information. This allows the system to provide up-to-date, near-real-time information on the health status of the cow. Second, the application has a scheduled process for updating classification results. This includes periodic checks and automated procedures that collect data from IoT tools and electronic health records. The collected data are then processed through the classification model to generate the latest health status. This ensures that even if users are not actively updating their information, the system can still provide up-to-date classification results based on the latest available data. Although the model is in the testing stage, it must be maintained over time to ensure that it remains accurate and up to date. This includes monitoring model performance over time, retraining the model if necessary, and updating model parameters and algorithms as new data become available.

## 4. Discussion

Various systems have been developed for monitoring the health status of animals. Some systems classify the state of health based on a database of accelerometers and gyroscopes, through which the current behavior of the animal is determined and, in the presence of deviations from pre-defined patterns [30], the presence or absence of a health problem is established. Other systems also perform health monitoring through automated visual observation [31], but rigorous and sustained exposure is limited due to time constraints and a lack of human resources [32]. Making use of the IoT enables monitoring certain parameters or health symptoms such as udder health, estrus events, feet and leg health, temperature, humidity, heart rate, rumination rate, and metabolic health [33,34].

Some non-invasive approaches [35] and technologies, such as visible video (VisV) and infrared thermal imagery (IRTI) [18], together with ML methods are aimed at animal health monitoring [14], identifying individual sick animals by observing eating and drinking behavior [36,37], temperature analysis of cold and heat stress [38,39], and video-based individual cow identification [40,41].

The examination of micro- and macroenvironmental factors would help farmers detect welfare problems at an early stage [42]. The physical conditions surrounding the microenvironment are defined as the macroenvironment (barn or pasture) [43]. Environmental temperature and relative humidity have been reported to influence the quantity and quality of milk [44,45,46,47] in dairy cows. Recently, intensive research has been aimed at elucidating reasons that can lead to dairy cow cold or heat stress within farm buildings. One such large-scale project was launched in April 2023 to understand the interaction between temperature, “microclimates” within farm buildings, and cow physiology and behavior [48]. The research [49] argues that when environmental conditions exceed a threshold limit that increases or decreases core body temperature, heat or cold stress occurs, and animal welfare can be compromised. Heat stress in lactating and dry cows can be assessed with animal- or environmental-based indicators [50]. Environmental modification can reduce heat accumulation and increase heat dissipation to protect calves and heifers from heat stress [51]. In [52], it is emphasized that the level of heat stress of all animals on a farm cannot be categorized by the same threshold, since animals have diverse biological characteristics (age, genotype, and production level) and the thermal state of animals is highly dependent on the specific environmental conditions each animal is exposed to.

This study is focused of the development of an integrated automated healthy classification system to study unexplored issues. Digitization is already an integral part of cows’ health monitoring [53], as it is convenient for optimizing costs associated with a large number of employees, reducing the probability of human error and potential losses from untimely responses to possible cows’ illness conditions. Detected deviations help vets by showing where to focus attention and act.

In this research, data are extracted from various sources such as non-invasive IoT devices and tools, weather stations, and health diaries and combined into a unified dataset. This approach differs from other studies like, for example [54], because an investigation is made whether and how micro- and macroenvironments influence cow health status. The study is then refined by adding individual information for each cow. This is important because each cow, depending on its age, lactation, and other specific factors, has a different endurance threshold. That is a major reason for why cows raised under the same conditions might exhibit different illness symptoms. The created combined and unified dataset allowed more accurate subsequent health status classification.

The data were processed by the well-defined workflow shown in Figure 2, which includes a sequence of processes for IoT data preparation, data modeling, and data integration support that streamlines the process and facilitates collaboration by providing a common framework for discussing data-processing steps. Furthermore, by following a workflow, the results can be reproduced and validated, which is important for ensuring the accuracy and reliability of the results. The presented workflow can only be used for classification and regression-type tasks. This is in line with the latest EU study [55], which sets a key objective in the use of data-driven technology for monitoring farm animals as being cost-effective and as robust as possible.

For numerical features shown in Figure 4 and objects, shown in Figure 5, the data analyses are performed separately. Also, a descriptive statistical analysis is provided, and the results of it, shown in Table 3, are used to examine the feature characteristics in more detail. Different types of distribution, scale, and data types along with 6.9% missing values and outliers with unusually high extreme values for some features are identified. In the case of IoT data, missing values are due to device problems or internet connection issues, which are common in IoT ecosystems, and outliers can be caused by a variety of factors, including sensor data errors, sensor calibration issues, or unexpected environmental conditions.

The Pearson method is used to create a correlation matrix between every two pairs of features, which is presented in Figure 6. The matrix showed a positive correlation of 56%, followed by 54% and 34%, and a negative correlation of 57%, 41%, and 27%. The analysis complied with the research results from [44,45] on the influence of environmental housing conditions on the milk yield of dairy cows. Multicollinearity is identified for micro and macro temperatures r = 0.97.

Based on the results obtained after the data analysis, several types of transformations are performed, such as casting to the correct data type, processing null, Na, NaN, and NaT values, processing outliers, feature engineering, balancing, standardization, and encoding. After the data transformations, the correct types of values are assigned, numerical values are cleaned of outliers and then scaled, and categorical variables are balanced and encoded with numerical values of 0 or 1.

Machine learning algorithms are a promising tool for improving farmers’ decision making [56]. In the modeling stage, five selected machine learning algorithms for classification are considered—K-nearest neighbor classifier, Gaussian naïve Bayes, XGBoost, and random forest classifier, which are presented in Table 2. The scikit-learn framework is used for their implementation. During the training of the algorithms, the grid search CV method is used (further tuned to improve the recall evaluation metric), with which the best hyperparameters are found for each individual algorithm. A disadvantage of this method is the long calculation time until acceptable results are obtained. Accuracy, recall, precision, and F1-score were used as models’ evaluation metrics. The important metric for this research is recall (sensitivity), due to the health check specificity. The models’ performance is presented in Table 4. All metrics refer to the test set. The random forest classifier was found to be the best-performing model with results such as 0.959 accuracy, 0.954 recall, and 0.97 precision with tuned hyperparameters for the entropy criterion, a maximum size of three random subsets of features to consider when splitting a node, minimum samples of ten leaves, fitting and classifying, and a total number of trees in the forest of three hundred.

A confusion matrix is created and presented in Figure 7, which shows the errors made by the model in percentage and quantitative values. The evaluation metric values are presented in Table 5, which shows the performance for each of the three classes and the accuracy of the model on training and testing data. The difference between train and test scores did not exceed 5%, which was the reason to consider that there was no overfitting at this stage.

The error analysis is made and the results show that there are three main contributors to model errors. The first factor is the presence of high amounts of daily milk yield—exceeding the values of 75% calculated in the statistical descriptive analysis. Another factor leading to classification errors is young cows. The third factor is the lactation phase—after lactation 4, the cow enters the mid-lactation cycle, where several changes occur, such as a decrease in the amount of milk, an increase in weight, and others. The features that are responsible for these factors need an additional amount of data and other related features.

The performance of any classification algorithm may depend on a range of factors, as discussed in [57]. The developed classification model is limited to the health status classification of milk cows.

There is a lot of variation in how animals are raised. However, the development of ML tools requires a degree of uniformity to achieve successful implementation. The data-driven tools work best on larger farms where standardized layouts are implemented [50].

The developed custom machine learning model is integrated into the cloud-based monitoring system as a new functionality. Cloud computing offers the advantages of centralized systems in terms of implementation, maintenance, updating, and use of unified logic [58]. Data from various IoT devices are transferred and analyzed in a cloud by using a single monitoring system. This is of great importance when a large set of IoT tools is operated.

Amazon services with different functions are selected, grouped, built, connected, and used for their operation, updating, and maintenance. Each of these services performs a specific task. The S3 buckets are responsible for the storage and synchronization of the data and the model. The AWS Glue workflow takes care of the extract, transform, load (ETL) processes. Step functions are responsible for application workflow as a series of event-driven steps. SageMaker is responsible for building and training a machine learning model and then directly deploying it into a production-ready hosted environment. The communication between the services is presented in Figure 3. The combination of all used AWS serverless services makes the newly built functionality available, reliable, and secure. But the services have the following operating limits:*S3 buckets* are available only for chosen AWS regions. There is no max bucket size or limit to the number of objects in it. The allowed operations in the bucket are “get”, “put”, and “list”.Glue: the total number of jobs, crawlers, and triggers within a workflow is up to 100.*Step functions* place quotas on the sizes of certain state machine parameters, such as the number of API actions during a certain time or the number of state machines that one can define (max request size limit of 256 KB).*SageMaker*. For an endpoint, the maximum size of the input data per invocation is limited to 6 MB. This value cannot be adjusted. For batch transform, the maximum size of the input data per invocation is 100 MB. This value cannot be adjusted.

The proposed solution aligns with the objective of promoting a sustainable and eco-friendly environment. AWS Cloud provides options for green computing and reducing carbon footprints. By strategically locating remote servers in regions with cooler climates and abundant water resources, the costs associated with server cooling can be substantially reduced, resulting in a significant decrease in environmental impact.

The obtained results are visualized in tabular form in the created client app interface presented in Figure 8. Classifications in it are updated in real time upon user request and on a schedule three times a day. Now, it serves to validate the classified results and the actual ones from veterinarians and farms. After passing the validation phase, the classified health status will be fed into an automated pipeline and in the case of a suspect or unhealthy status, farmers will be notified immediately.

In our future work, we plan to enrich the current dataset with cow diagnosis when identifying unhealthy status. There are also plans to incorporate data analysis of milk components, including protein, fat, and lactose, for each individual cow. This additional data will enable us to detect and identify health issues such as mastitis, subclinical ketosis, acidosis, and feeding problems. Furthermore, we aim to integrate observations of changes in cow behavior and activity patterns to further enhance and complement the results of our research. By combining these various sources of information, we can establish a robust framework for assessing the health of dairy cows as part of a precision breeding approach. These findings will serve as valuable indicators for monitoring and improving the overall health and well-being of dairy herds.

## 5. Conclusions

This research proposes an approach to combine and process data from different sources related to dairy cows to train an algorithm and create a model that manages to summarize the data, rather than just memorize them. The presented systematized workflow is followed, and it includes methods and approaches for processing IoT data to develop and integrate a robust, efficient, and reusable flow for data analysis, processing, and modeling.

The experimental phase includes univariate and bivariate data analyses. The results clearly indicate the collected data are heterogeneous and the presence of outliers and missing and duplicate values. Correlation analysis identifies both positive and negative correlations between features and measures the significance of the parameters for the specified problem. Multicollinearity is identified for some features. The analyses highlight the need for subsequent data-pre-processing steps that include missing value imputation, outlier correction, class balancing, data normalization, and feature encoding.

Five machine learning algorithms are trained, and hyperparameters are fine-tuned using the grid search method. The random forest classifier is the best-performing model with results such as 0.959 accuracy, 0.954 recall (sensitivity), and 0.97 precision. The trained model accurately classifies the health status of cows into three categories.

Subsequent error analysis is performed, and results emphasize the need for additional data and highlight the inherent complexity of biological systems, where identical conditions can lead to different outcomes.

Amazon Web Services is used for the development and deployment of the proposed approach. A dedicated architectural design is created for integrating new data and a machine learning model training in a cloud environment using serverless cloud services. A group of AWS serverless services are selected from over 200 others. The authors’ approach is used for services’ organization, configuration, inter-service communication methods, and system roles. This provides a practical solution for integrating and using data and models in a scalable and cost-effective way, which is particularly suitable for the livestock industry.

## Figures and Tables

**Figure 1 animals-13-03254-f001:**
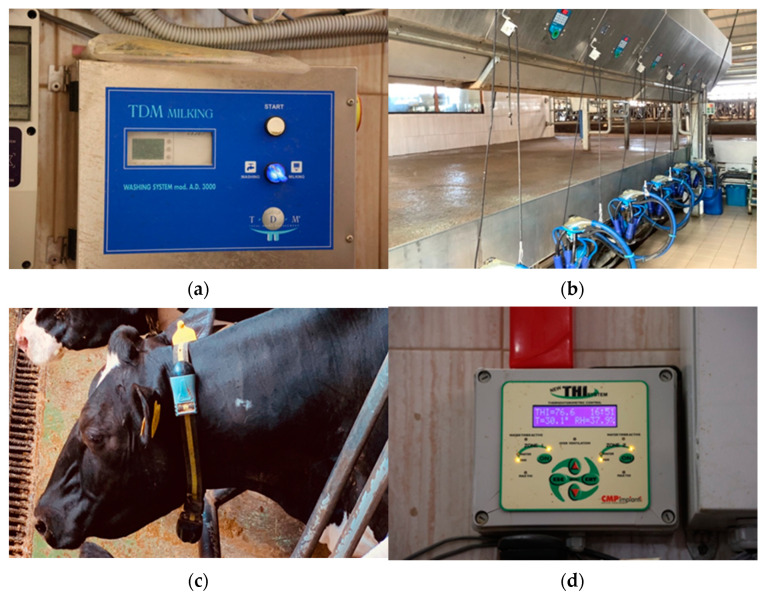
(**a**) An automated milking and washing system; (**b**) automated milking device with a sensor for correct placement and a counter for milk pumped; (**c**) cow’s neck IoT device with sensors for temperature, humidity, and animal identification; (**d**) a barn environment management system measures temperature (T), humidity (RH), and temperature humidity index (THI).

**Figure 2 animals-13-03254-f002:**
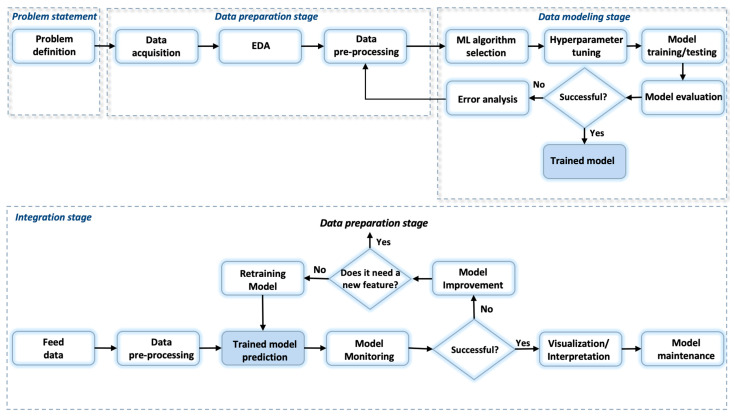
Workflow for IoT data processing, modelling, and machine learning model integration in a production environment.

**Figure 3 animals-13-03254-f003:**
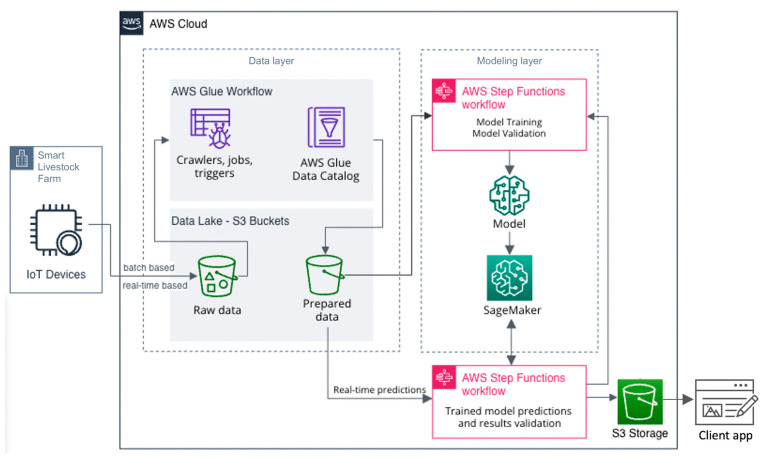
New data and trained model integration architecture diagram.

**Figure 4 animals-13-03254-f004:**
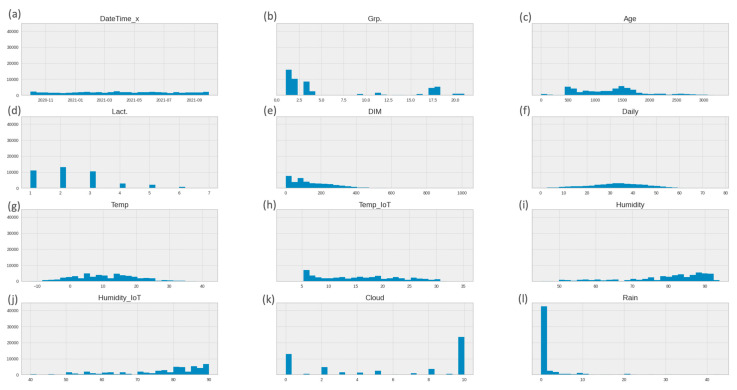
Data distribution histograms of numerical features. (**a**) “DataTime_x” data distribution is shown by the *x*-axis timestamp, and *y*-axis counts samples; (**b**) “Grp.” data distribution is shown by the *x*-axis number of group locations, and *y*-axis counts samples; (**c**) “Age” data distribution is shown by the *x*-axis cow age (in days), and the *y*-axis counts samples; (**d**) “Lact.” data distribution is shown by the *x*-axis cow lactation phase, and the *y*-axis counts samples; (**e**) “DIM” data distribution is shown by the *x*-axis number of days in milk (in liters), and the *y*-axis counts samples; (**f**) “Daily Milk” data distribution is shown by the *x*-axis quantity of milk per day, and the *y*-axis counts samples; (**g**) “Temp” data distribution is shown by the *x*-axis degrees (in Celsius), and the *y*-axis counts samples; (**h**) “Temp_IoT” data distribution is shown by the *x*-axis degrees (in Celsius), and the *y*-axis counts samples; (**i**) “Humidity” data distribution is shown by the *x*-axis humidity level (in percentage), and the *y*-axis counts samples; (**j**) “Humidity_IoT” data distribution is shown by the *x*-axis humidity level (in percentage), and the *y*-axis counts samples; (**k**) “Cloud” data distribution is shown by the *x*-axis cloudiness level index, and the *y*-axis counts samples; (**l**) “Rain” data distribution is shown by the *x*-axis quantity of rain (liters per square meter), and the *y*-axis counts samples.

**Figure 5 animals-13-03254-f005:**
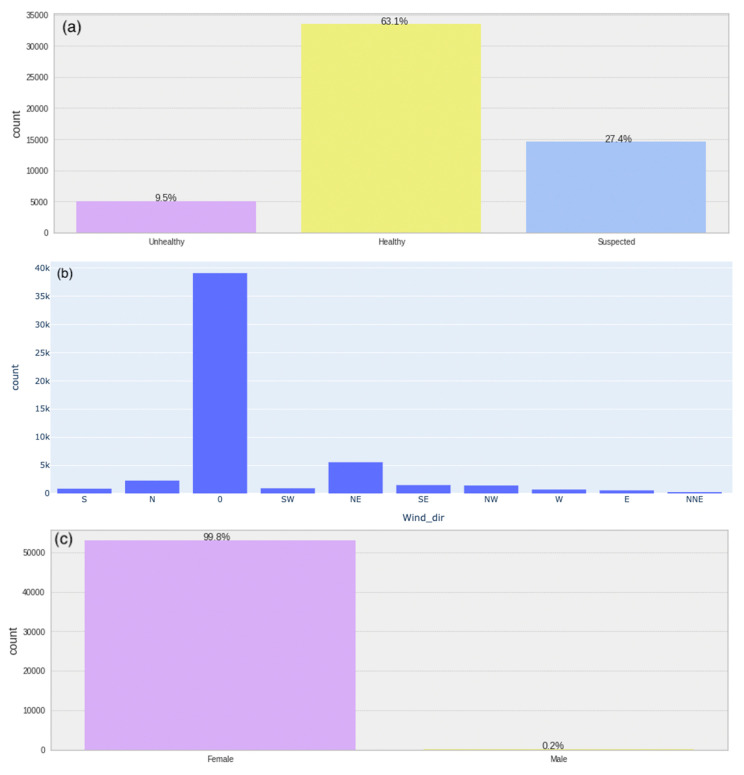
Data distribution of categorical features: (**a**) health status categorical distribution: unhealthy 9.5%, healthy: 63.1%, and suspect: 27.4%; (**b**) wind direction categorical distribution; (**c**) gender categorical distribution.

**Figure 6 animals-13-03254-f006:**
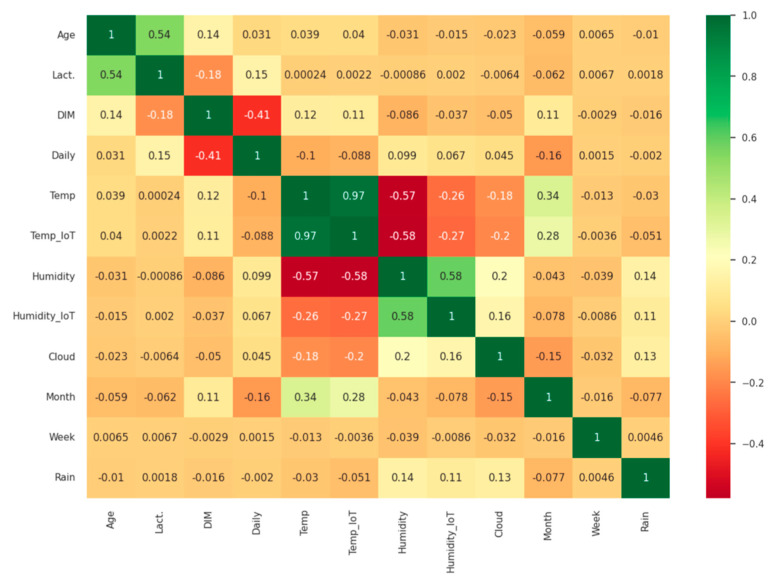
Correlation heatmap. The rows represent the relationship between each pair of variables. The values in the cells indicate the strength of the relationship, with positive values indicating a positive relationship and negative values indicating a negative relationship. The color coding of the cells identifies relationships between variables. The green color represents a strong relationship (values are near *1*). The red color represents a negative relationship (values are near −1). The yellow color represents no correlation (values are near *0*).

**Figure 7 animals-13-03254-f007:**
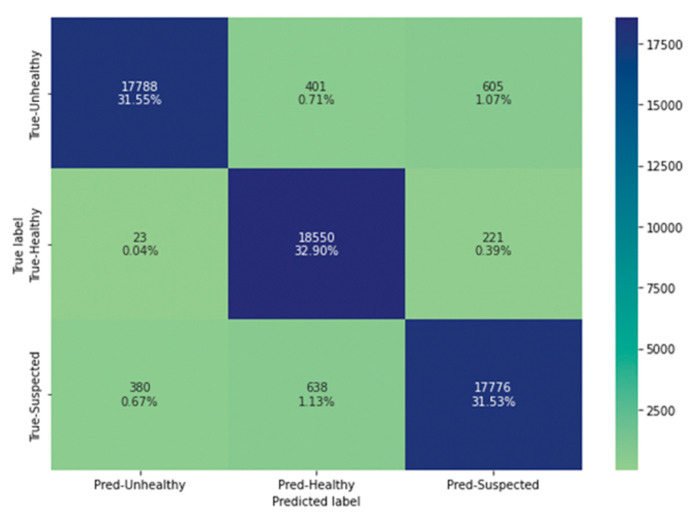
Confusion Matrix of Random Forest Classifier model.

**Figure 8 animals-13-03254-f008:**
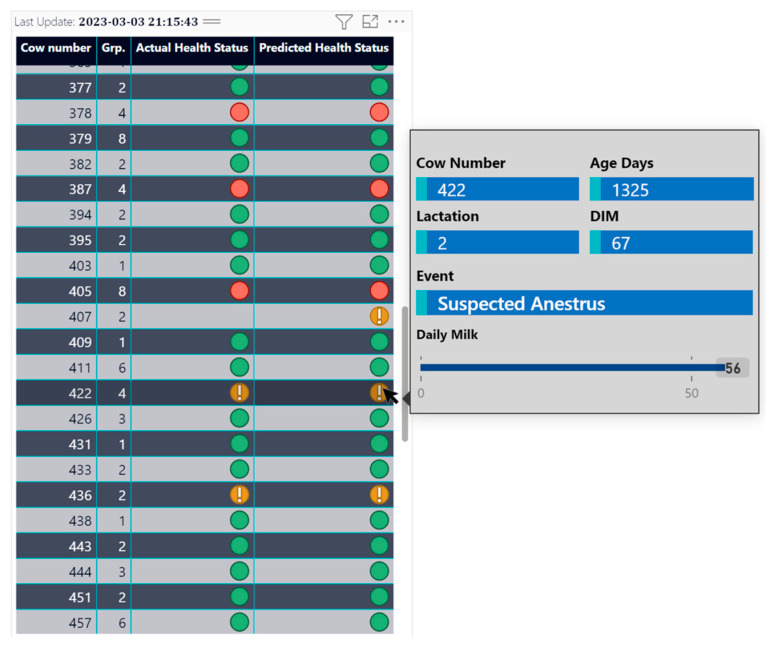
A preview of the table representing the health status classification of the cows. The table shows values for time, cow’s number (ID), cow’s current group, actual and predicted health status. There are three health status types. Healthy cows are indicated with a green point, suspect cows are indicated with a yellow point, and unhealthy cows have a red point. A tooltip is activated when the cursor hovers over a corresponding line. It shows additional information for the cow’s number, age, lactation, DIM, event, and amount of milk.

**Table 1 animals-13-03254-t001:** Features and their meaning.

Feature	Feature Description
Cow	Registration number of the cow (ID)
Age	Age of the cow in days
Gender	Gender of the cow
Lact.	Lactation—breastfeeding period
DIM	The number of days the average milking cow in the herd is milking from calving day
Daily Milk	Average daily amount of milk
Grp.	The group in which the cow is located
Temp.	Temperature of cow’s microclimate
Humidity Temp_IoT Humidity_IoT	Humidity of cow’s microclimate Temperature of cow’s macroclimate Humidity of cow’s macroclimate
Wind_dir	Wind direction outside
Cloud	Degree of cloudiness outside
Rain	Amount of precipitation (liters per square meter)
Data_time	Timestamp of measurements
Healthy	The health status of the animal data

**Table 2 animals-13-03254-t002:** Machine learning algorithms selected for the research.

ML Algorithm	Algorithm Description
K-nearest neighbors (KNN)	It is a non-parametric algorithm. The rationale of the K-nearest neighbors’ algorithm is that each sample can be represented by K-nearest neighbors. The distance between the test and training samples can calculate the classification for the samples. This distance usually uses “Euclidean distance” or “Manhattan distance” [23]. According to the distance order, the test sample is classified as the closest class to it. To measure the sampling distance on the same scale, all features are normalized and then the K-value selection is obtained by cross-validation.
Gaussian naïve Bayes (GNB)	This is a probabilistic algorithm based on the Bayes theorem, where one of the hypotheses is the assumption of strong independence between features. It supports continuous-valued features and models each one as conforming to a Gaussian distribution.
Decision tree classifier (DTC)	It is a non-parametric algorithm where the data are continuously divided into smaller parts until it reaches a class or is truncated by a hyperparameter such as *max_depth*. It has a hierarchical tree structure, which consists of a root node (represents a feature), decision nodes which represent the logic statements used to split or divide the data into two parts, and leaf nodes (represent the outcome).
XGBoost (XGB)	It is a boosting ensemble algorithm that implements the gradient-boosting decision tree algorithm. It trains individual decision trees sequentially, with each tree trying to classify the errors of the previous tree. The model combines the classifications of all the individual trees to make a final classification.
Random forest classifier (RFC)	This ensemble algorithm is a combination of multiple decision trees. Each tree is built independently using bootstrap resampling with replacement replicates of the features and samples, which creates independence in each dataset used to create the next tree in the forest allowing the random forest model to give a robust classification. The classification is made by taking the majority vote on the classifications made by each decision tree. This improves classification accuracy and generalizability while avoiding overfitting of the model...

**Table 3 animals-13-03254-t003:** Descriptive statistical analysis report of numerical features.

Statistical Measures	Grp.	Age	Lact.	DIM	Daily Milk	Temp.	Temp_IoT	Humidity	Humidity_IoT	Cloudiness	Rain
Number of records	53,155	53,155	40,859	40,859	37,408	53,155	53,057	53,155	53,057	53,155	53,155
Mean	6.266	1315.875	2.38	163.27	33.11	11.05	15.84	79.42	75.90	5.911	1.953
std	6.903	641.805	1.236	145.57	13.11	9.48	7.41	11.24	12.44	4.293	5.49
Min	1	0	1	0	0.2	−12	2	45	40	0	0
25%	1	792	1	61	24.6	4.2	9	74	68	1	0
50%	3	1353	2	123	33.7	10.8	15.8	83	80	8	0
75%	12	1641	3	239	42.4	17.8	22	88	86	10	0.30
Max	21	3293	7	1013	77.4	42.0	35.2	94	90	10	43

**Table 4 animals-13-03254-t004:** Models’ performance.

Machine Learning Algorithm	Hyperparameters	Recall	Precision	Accuracy
K-nearest neighbor classifier (KNN)	k-neighbors = 11 weights = uniform			
	p = 2	0.726	0.719	0.816
	n_jobs = −1			
Gaussian naïve Bayes (GNB)	var_smoothing = 0.035111	0.528	0.532	0.620
Decision tree classifier (DTC)	max_features = 10	0.632	0.740	0.828
	min_samples_leaf = 10			
	min_samples_split = 3			
	criterion = gini			
	n_jobs = −1			
XGBoost (XGB)	colsample_bytree = 0.75	0.609	0.70	0.784
	early_stopping_rounds = 10			
	learning_rate = 0.1			
	max_depth = 2			
	min_child_weight = 1			
	subsample = 0.75			
	n_estimators = 100			
	n_jobs = −1			
Random forest classifier (RFC)	bootstrap = False	0.954	0.97	0.954
	criterion = entropy			
	max_features = 3			
	min_samples_leaf = 10			
	n_estimators = 300			
	verbose = 1			

**Table 5 animals-13-03254-t005:** Random Forest Classifier Evaluation Report.

Class	Precision	Recall	F1-Score
0	0.98	0.95	0.96
1	0.95	0.99	0.97
2	0.96	0.95	0.95
Accuracy			Test set 0.9598 Training set 0.9620
Number of records			56,382

## Data Availability

The data presented in this study are available on request from the corresponding author.

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
