# Peer review of "Health Status Classification for Cows Using Machine Learning and Data Management on AWS Cloud"

_animals, 2023, doi:10.3390/ani13203254_

Round 1

Reviewer 1 Report

Dear authors,

Many thanks for this piece of work. I read it with interest and I found it meritorious of being considered further to publication.

I reckon that the manuscript would benefit of minor revision. 

The manuscript is well structured. The title is concise and informative, the abstract stands alone and reports sufficient information . Also the summary fits the purpose to convey main findings to the reader.

Introduction  is too long and many of the references cited would be better used if moved to the discussion. Indeed, the introduction is limited in being self explanatory to sharpen background, hypothesis and objectives. Moreover, some important references were not instead cited by authors. I would warmly recommend authors to cite Cappai et al., "Long term performance of RFID technology...", Small Ruminant Research 2013, doi:10.1016/j.smallrumres.2013.12.031.

Material and methods are apprpirate to meet the goals, and clear and well organized to allow repetition.

Results are displayed adequately and discussion  may benefit of some concepts wrongly anticipated to introduction .

Conclusion are fine and report main findings.

Please, update reference and citations and provide changes to Introduction/discussion.

Thank you

Author Response

The authors would like to sincerely thank the reviewer for reviewing the manuscript and for the valuable comments.

We did our very best to address them in the revised version of the manuscript.

The authors thank you for the provided reference. We find it useful and relevant to our research.

The references and citations are updated. Changes to the Introduction/discussion are provided.

Reviewer 2 Report

This manuscript presented a scheme for classifying cow health status using machine learning models. It combined data from both micro and macro environmental monitoring of cows with specific information, and proposes a process that involves various data processing methods. This manuscript described the use of five different machine learning algorithms to monitor the health status of cows, and ultimately selects the Random Forest Classifier as the best performer. The trained model was integrated into cloud infrastructure services in the AWS environment, and a new interface was created in the client application to visualize the classification results of the machine learning model. This scheme could improve the accuracy and efficiency of classifying cow health status, and was beneficial to the sustainable development of the dairy industry. However, there were still some parts that need improvement:

1. The introduction content is rich but there is a lot of repetitive content. Please streamline the language. Additionally, it should be described in the introduction what application these five selected algorithms have in the animal husbandry field.

2. The content from Line 160-162 is repeated in the introduction and should be deleted.

3. The entire article is too long, which means descriptive and explanatory content should be reduced. For example, in 2.2.2, the description of data preparation is overly verbose.

4. In Table 2, please add references for each algorithm.

5. Insufficient description: Line-173, what data were collected through neck IoT device.

6. Lack of dataset size.

7. In the results section, a simple description of the data in the figures should be provided, while a detailed discussion should be included in the section

8. Discussion. there should not be too much background description in the discussion, but rather a deeper explanation and analysis of the results. The repetition and verbosity are the biggest issues in this manuscript. Please delete unnecessary descriptions or include them in the supplementary materials if they need to be added.

9. The conclusion should reduce statements about the research background, distill the research results, and list them in sections.

10. Can the optimal model obtained through this research be applied to other fields?

none

Author Response

The authors would like to sincerely thank the reviewer for reviewing the manuscript and for the valuable comments.

We did our very best to address them in the revised version of the manuscript.

  1. The introduction content is rich but there is a lot of repetitive content. Please streamline the language. Additionally, it should be described in the introduction what application these five selected algorithms have in the animal husbandry field.

Response 1:

A new text is added to the introduction section with the application of the five selected and used algorithms.

  1. The content from Line 160-162 is repeated in the introduction and should be deleted.

Response 2:

The authors thank for the remark. Duplicates are removed.

  1. The entire article is too long, which means descriptive and explanatory content should be reduced. For example, in 2.2.2, the description of data preparation is overly verbose.

Response 3:

Explanatory content is reduced. Overly verbose paragraphs are revised. Changes are made.

  1. In Table 2, please add references for each algorithm.

Response 4:

Reference is added in the manuscript.

  1. Insufficient description: Line-173, what data were collected through neck IoT device.

Response 5:

The authors thank for the remark.

An additional description of collected data from the neck IoT device is added in section 2.1 Data collection (figure 1 (c)) and 2.2 Data preparation.

  1. Lack of dataset size.

Response 6:

The authors thank for the remark.

The missed information is added in section 2.1 Data collection.

  1. In the results section, a simple description of the data in the figures should be provided, while a detailed discussion should be included in the section.

Response 7:

Changes are made to the manuscript in Section 3: Results and in Section 4: Discussion

  1. there should not be too much background description in the discussion, but rather a deeper explanation and analysis of the results. The repetition and verbosity are the biggest issues in this manuscript. Please delete unnecessary descriptions or include them in the supplementary materials if they need to be added.

Response 8:

Unnecessary descriptions are removed from the manuscript.

  1. The conclusion should reduce statements about the research background, distill the research results, and list them in sections.

Response 9:

 Changes are made in the conclusion section.

  1. Can the optimal model obtained through this research be applied to other fields?

Response 10:

The article primarily focuses on the cows’ health status classification. In fact, the overall approach and resulting optimal model can be applied to other ruminants, e.g. sheep as long as the model is retrained using adequate and relevant data and optimal hyper-parameters subsequently used.

Beyond livestock, the strategies employed could be used for predictive health assessments in humans, especially when wearable tech and environmental data are combined.

However, while the methodologies and strategies can potentially be applied to other fields, several domain-specific challenges and modifications should be addressed. Each application will have its own unique features, data peculiarities, and domain-specific nuances that need consideration. The feature importance, data distribution, and potential anomalies described for cow health might not be directly applicable to another domain.

Reviewer 3 Report

The manuscript is voluminous and confusing due to poorly broken tables that are difficult to follow, as well as bulky images. In this work, it is necessary to examine the difference in the value of the investigated parameters in sick and healthy cows, as well as in function of the type of disease diagnosed in the cows. It is necessary to examine the difference in correlation matrices in healthy and diseased cows, as well as by type of disease. This is valuable information for researchers in the field of veterinary medicine. In the paper, the use of this database and technology should be validated so that after the classification is completed, the veterinarian will access additional examination and therapy, so it is necessary to include data on subsequent diagnosis and therapy and their effect in this manuscript, in order to be sure that this application/base really has practical value. Without that, everything remains at the level of statistical speculation with a large amount of data. In short: simplify presentation and include veterinary treatment after classification based on this software.

Author Response

The authors would like to thank the reviewer for the observations.

  1. The manuscript is voluminous and confusing due to poorly broken tables that are difficult to follow, as well as bulky images.

Response 1:

The authors would like to sincerely apologize for the inconvenience.

Following the MDPI rules - Instructions for Authors section "Preparation of figures, schematics and tables", authors will provide editors with all figures in full size and quality (minimum 1000 pixels width/height or resolution 300 dpi or higher) in a single zip archive. This will allow readers to see the original quality of all figures.

  1. In this work, it is necessary to examine the difference in the value of the investigated parameters in sick and healthy cows, as well as in function of the type of disease diagnosed in the cows. It is necessary to examine the difference in correlation matrices in healthy and diseased cows, as well as by type of disease. This is valuable information for researchers in the field of veterinary medicine.

Response 2:

The authors agree with the reviewer's suggestion, but they think it must be done in a separate investigation where additional data containing more diseased animals will be included and used to examine and interpret the influence of individual parameters with greater accuracy. Additionally, the parameter changes and correlations related to a certain diagnosis will be investigated.

  1. In the paper, the use of this database and technology should be validated so that after the classification is completed, the veterinarian will access additional examination and therapy, so it is necessary to include data on subsequent diagnosis and therapy and their effect in this manuscript, in order to be sure that this application/base really has practical value. Without that, everything remains at the level of statistical speculation with a large amount of data.

In short: simplify presentation and include veterinary treatment after classification based on this software.

Response 3:

The authors state In Section 3.4 that the model is under validation. The farm where the solution is integrated successfully validates the obtained results. Moreover, in case of confirmation that the cow is unhealthy, they record the diagnosis of the animal (figure 8 - tooltip). This helps us to gather more focused observations which will be used to enrich the current dataset and, in a future study, to be able to upgrade the model built in this study by classifying the diagnosis when the status of the cow is established as unhealthy. The authors would like to mention that gathering such additional observations is a relatively slow process as it requires manual data input from vets and an extra level of validation which unfortunately makes the automation of the process quite cumbersome currently. Steps to overcome such limitations will be also included in future.

The presentation was simplified. The inclusion of veterinary treatment after classification was added as future work.

Round 2

Reviewer 3 Report

Not all recommendations were fully followed. The sensitivity of the system and the correlation between parameters in healthy and diseased cows are not visible.

Author Response

The authors apologize.

The Sensitivity (recall) is added to the text. The correlations between features in healthy and diseased cows are added as a new Appendix C.